# MSF-Net: A Lightweight Multi-Scale Feature Fusion Network for Skin Lesion Segmentation

**DOI:** 10.3390/biomedicines11061733

**Published:** 2023-06-16

**Authors:** Dangguo Shao, Lifan Ren, Lei Ma

**Affiliations:** 1Faculty of Information Engineering and Automation, Kunming University of Science and Technology, Kunming 650500, China; shaodg@kust.edu.cn (D.S.); 20212104013@stu.kust.edu.cn (L.R.); 2Yunnan Key Laboratory of Artificial Intelligence, Kunming University of Science and Technology, Kunming 650500, China

**Keywords:** skin lesion segmentation, attentional mechanism, dilated convolution, multi-scale feature fusion

## Abstract

Segmentation of skin lesion images facilitates the early diagnosis of melanoma. However, this remains a challenging task due to the diversity of target scales, irregular segmentation shapes, low contrast, and blurred boundaries of dermatological graphics. This paper proposes a multi-scale feature fusion network (MSF-Net) based on comprehensive attention convolutional neural network (CA-Net). We introduce the spatial attention mechanism in the convolution block through the residual connection to focus on the key regions. Meanwhile, Multi-scale Dilated Convolution Modules (MDC) and Multi-scale Feature Fusion Modules (MFF) are introduced to extract context information across scales and adaptively adjust the receptive field size of the feature map. We conducted many experiments on the public data set ISIC2018 to verify the validity of MSF-Net. The ablation experiment demonstrated the effectiveness of our three modules. The comparison experiment with the existing advanced network confirms that MSF-Net can achieve better segmentation under fewer parameters.

## 1. Introduction

Skin cancer is a common type of malignancy of which melanoma is one of the deadliest skin lesions [1]. According to the American Skin Cancer Society report, melanoma has become one of the most common problems worldwide. In terms of pricing, it is estimated that $3.3 billion a year is spent on treating skin cancer. In addition, melanoma has attracted increasing attention from clinicians and researchers, who emphasize that early detection of melanoma can significantly reduce mortality [2].

Skin diseases are mainly detected by dermoscopy technology [3]. Since the diagnostic process of dermatology is tedious and partly subjective to the physician’s opinion, computer-aided diagnosis (CAD) has its value [4], and image segmentation of dermatology is an integral part. However, this remains challenging due to the diversity of target scales in dermatological images and the problems of irregular segmentation shapes, low contrast, and blurred boundaries. The specific example is shown in Figure 1. The ground truth in the figure is a professional physician’s manual annotation of the lesioned region. Our research task belongs to the binary segmentation task, where annotates the lesioned region as one and the non-lesioned region as zero. These pathological images lack global semantic and context information guidance, which can easily lead to wrong segmentation results. To further address these challenges, we propose the MSF-Net based on comprehensive attention convolutional neural network (CA-Net) [5] while combining multi-scale feature fusion methods. There are differences in the scale of the lesion area. Large object detection is better at low resolution with a global field of perception. Small objects are better predicted at high resolution, which preserves some details such as edge information. Therefore, Multi-scale feature fusion remarkably affects medical skin lesion image segmentation.

This paper proposes three modules, the S-conv Blocks, the Multi-scale Dilated Convolution Modules (MDC), and the Multi-scale Feature Fusion Modules (MFF). These modules are plug-and-play, and many ablation results show that each module improves the segmentation accuracy of the network compared to CA-Net, which is of great significance for the practical clinical application of melanoma segmentation. To sum up, our research achievements mainly include the following aspects:The spatial attention mechanism is introduced into the convolution modules to adjust the weight of the image pixel, focus on crucial information, and suppress irrelevant information.Relevant information of different scales can be obtained through the parallel branch of four dilated convolutions, and the receptive field size of the feature map of different scales can be adjusted adaptively according to the input information of different scales.The MFF modules are proposed to fuse relevant information of multiple scales.

## 2. Related Works

### 2.1. Image Segmentation

According to different methods, traditional image segmentation [6] can be roughly divided into two directions: edge detection and region growth. The former performs segmentation based on the discontinuity between the area of interest and its adjacent areas. The latter realizes image segmentation by local similarity. However, these methods have many shortcomings such as low segmentation accuracy, poor anti-noise ability and robustness, and manual feature extraction.

Several achievements have been made in applying convolutional neural networks (CNNs) to image segmentation in recent years. Long et al., 2015 [7] have proposed a fully convolutional neural network (FCN) to solve the problem of semantic segmentation, which abandons the entire connection layer in CNN and chooses to use the convolutional layer instead and upsamples the final output to the size of the original image so that the production is no longer a possible value. However, FCN has an obvious shortcoming, which needs to consider the spatial position and context information of pixels, causing damage to local and global information.

With the development of image segmentation, more and more researchers are committed to using image segmentation in the field of medical images to assist in the diagnosis of some critical diseases including liver cancer [8], lung cancer [9], prostate cancer [10], and some kidney diseases [11]. Ronneberger et al., 2015 [12] borrowed the ideas of FCN and then proposed an end-to-end “U-shaped” network, U-Net, for medical image segmentation. U-Net fuses different levels of information through remote hopping connections to compensate for some feature loss caused by FCN. Many excellent networks have subsequently emerged based on this network, including AttU-Net [13], Unet++ [14], Unet 3+ [15], R2U-Net [16], Dense U-Net [17], etc. On this basis, Ran Gu et al., 2021 proposed a comprehensive attention convolutional neural network, CA-Net [5].

### 2.2. Attention Mechanism

Oktay et al. [13] recently added a spatial attention mechanism to the jump-connected part of U-Net for segmenting the pancreas from abdominal CT images. Roy et al. [18] proposed a “Squeeze and Excite” (scSE) framework of parallel spaces and channels for the segmentation of multiple organs in the whole brain and abdomen. Kaul et al., 2019 [19] combined the SE module with ResNet to propose Focusnet to segment melanoma and lung lesions. Xiang Li et al., 2019 [20] fused channel attention and dilated convolution to propose SK-Net, which can adaptively select a suitable convolution kernel size.

### 2.3. Multi-Scale Feature Fusion

Low-resolution features have sufficient semantic information, while high-resolution parts contain rich spatial information and little global semantic contextual information. Therefore, the segmentation accuracy can be significantly improved by fully integrating low-resolution and high-resolution features.

To improve the adaptability of the network, Szegedy et al., 2014 [21] proposed the Inception v1 module, which uses convolutional kernels of different scale sizes at the channel level to extract information. Subsequently, Inception v2 and v3 modules [22] (2015) were proposed to reduce network parameters and enhance the non-linear expression ability of the network. To reduce the extraction of relevant repetitive features, He et al., 2015 [23] use pooling layers with different window sizes to extract features at different scales. The feature pyramid uses shallow features to separate simple objects and deep features to separate complex things. Chen et al., 2017 [24], inspired by the SPP block, proposed the atrous spatial pyramid pooling module (ASPP) in combination with the idea of atrous convolution. At the same level, the network uses the convolution kernel of multiple scales generated by the atrous convolution to extract information. It integrates the extracted features to form the input of the next layer.

## 3. Methods

The model framework of MSF-Net is shown in Figure 2, which consists of the following components: The S-conv blocks, the MFF modules, and the MDC modules.

Firstly, the segmented image is sent to the encoding network. S-conv blocks are used to replace the convolution blocks in the traditional U-Net.

Secondly, the MDC modules are added after the convolution modules and pooling, which can use appropriate background information to segment the diseased areas in the image. In contrast to the constant receptive field of ordinary CNNs, the MDC module mimics the neurons in the human visual cortex. It continuously adjusts the receptive field size of its neurons according to the external stimuli.

Thirdly, this paper uses MFF modules in both the encoder and decoder to extract and fuse multi-scale relevant information. Convolution kernels at different scales are used to process input information so that features of different scales can be combined to locate the lesion area more accurately. At the same time, this paper connects the encoder and decoder in the MFF modules, integrates high- and low-level semantic information, and further enhances the segmentation effect.

### 3.1. S-Conv Block

The traditional convolution module of U net is two 3 × 3 convolutions, and successive convolutions will miss a lot of detailed information. Therefore, the S-conv module replaces the convolution module of the encoder part. The frame diagram of S-conv is shown in Figure 3.

Our convolution module consists of 1 × 1 convolution, 3 × 3 convolution, Batch Normalization (Bn), and Relu layer. The S-conv module has two branches, one of which is two consecutive 3 × 3 convolution blocks to extract features, and the output feature map is ${Yr}_{1}\in R^{H\times W\times C_{in}}$. The other branch takes a 1 × 1 convolution to reduce the number of channels to 1, and then the input feature graph is ${Yr}_{2}\in R^{H\times W}$. Afterwards, a sigmoid function scales the values of the trait map to between [0, 1], which results in an attention coefficient map. Finally, the obtained attention coefficient graph is multiplied by the output feature graph of the convolution layer to calibrate the input and locate the critical information. Then, the final output ${Yr}_{out}$ of the S-conv module can be written as
(1)$${Yr}_{out}={Yr}_{1}*S\left({Yr}_{2}\right)$$
where ${Yr}_{1}$ is the output of the two 3 × 3 convolution layers, ${Yr}_{2}$ is the output of the 1 × 1 convolution layer, and *S* is the sigmoid function activation.

### 3.2. Multi-Scale Dilated Convolution Module

The traditional U-net framework contains a lot of convolutions and four pooling operations, so a lot of detailed information will be lost in the convolution and pooling process, especially the pooling operation. To compensate for this loss, we introduce the MDC module. It comprises two parts. The first one is the dilated convolution module. It can expand the receptive field without losing the resolution, which is superior to the traditional pooling operation. At the same time, weights are shared among our dilated convolution branches, which is a larger receptive field that can be obtained without introducing additional parameter numbers. It should be noted that the same parameter number can be trained on inputs of different scales. The second part is the fusion module (F module), which can automatically adjust the receptive field of the feature map of different scales to highlight some critical areas of other scales. The MDC module is inspired by the SK net [20] and ASPP [24] modules. The module uses a spatial attention mechanism on the fused information to generate two different attention maps applied to two branches. Additional focal points of these branches can generate various receptive fields of different sizes so that the receptive fields can be adjusted adaptively according to the input. The framework of MDC is shown in Figure 4.

First, the module enlarges the perceptual field of the input image by four dilated convolutions with different dilation rates. Here, the expansion rate r is set as 1, 2, 4, and 10 so that the receptive field (Rf) of the 3 × 3 convolution lernel can be expanded to
(2)$${Rf}_{3\times 3}\;(x)\;=\;2\;\times \;(r\;-\;1)\;+\;3$$

Therefore, when r = 2, 4, and 10, the receptive fields at this time are 5 × 5, 9 × 9, and 21 × 21, respectively. MDC is only used four times. The reason why this module is not adopted after the last max pooling is that after four times of maximum pooling, the feature map of the input is too small, only 14 × 18, which is less than the sensitivity field of the convolution kernel when r = 10. Taking the first MDC module as an example, the input feature map is $Y_{in}\in R^{H\times W\times C_{in}}$. After four dilated convolutions, we obtain four feature maps ($Y_{1}$, $Y_{2}$, $Y_{3}$, $Y_{4}$)∈$R^{H\times W\times C_{in}}$ respectively. We set the padding to 1, 2, 4, and 10 to keep the size of the output feature map the same as the input. The MDC module uses global high-level semantic information to quickly locate segmentation targets and local low-level spatial information to refine edge details.

After that, we fuse these different scales of feature information with the F module. Module F has two branches, taking $Y_{1}$ and $Y_{2}$ as example. First, the obtained feature maps $Y_{1}$ and $Y_{2}$ are joined together at the channel level to obtain feature map $Y_{12}^{\prime }$∈$R^{H\times W\times {2C}_{in}}$. Where H, W and C are the feature graph’s height, width and number of channels, respectively. The first branch passes through a 1 × 1 convolutional layer to reduce the dimensionality and change the number of channels to C. Then, it passes through a 3 × 3 convolutional layers with an output channel number of 2. At this point, we obtain the feature map $Y_{12}^{\prime \prime }$∈$R^{H\times W\times 2}$. We add batch normalization layers (BN) and ReLU activation layers to all convolutional layers. Then, after a SoftMax function, two different spatial attention coefficient maps $M_{a1}$ and $M_{a2}$ are obtained, and $M_{a1}$ and $M_{a2}$ are multiplied with $Y_{1}$ and $Y_{2}$ respectively to obtain $Y_{M1}$.
(3)$$Y_{M1}=Y_{1}*M_{a1}+Y_{2}*M_{a2}$$

The other branch passes through a 1 × 1 convolutional layer with the output channel number of 1 to obtain the feature map ${Y12}^{\prime \prime \prime }$. Then after a sigmoid function for activation, the activation value is mapped to the range [0, 1]. At this time, a spatial attention weight map $M_{a3}$ is obtained. $M_{a3}$ is multiplied with $Y_{12}$ to obtain $Y_{M2}$, which is used to recalibrate the spatial information, and finally, a 1 × 1 convolutional layer is passed to recover the number of channels.
(4)$$Y_{M2}=H\;[\;S\;(Y_{12}^{\prime \prime \prime })\;*Y_{12}^{\prime }]$$
where ∗ denotes the multiplication of elements, H is the number of channels halved, and S is the Sigmoid function activation.

Then, the final output of this layer can be written as
(5)$$Y_{12}=Y_{M1}+Y_{M2}$$

The calculation process of other branches is similar to the one above, and the final output can be written as
(6)$$Y_{out}=Y_{1234}+Y_{in}$$
where $Y_{1234}$ is the output feature map of the fused four layers, and here the residual connection is used to sum up with the input $Y_{in}$.

After the operation above, we can effectively fuse multi-scale features and adaptively adjust the perceptual field size of different scale feature maps.

### 3.3. Multi-Scale Feature Fusion Module

MFF modules are used in encoders and decoders to mine relevant information across scales. The module was inspired by Inception v2 [22]. In many cases, the features difficult to extract at the first scale are easy to capture at the other scale. Therefore, by combining the features of different scales, a lot of missing or hidden details can be obtained. The frame of the MFF module is shown in Figure 5.

The MFF module has three branches and outputs three feature maps $F_{1}$, $F_{2}$, $F_{3}$ ($F_{1}\in R^{H\times W\times C_{in}/4}$, $F_{2}\in R^{H\times W\times C_{in}/2}$, $F_{3}\in R^{H\times W\times C_{in}/4}$). All three branches are convolved by 1 × 1 to vary the number of channels and capture the lower-level features. The number of output channels of each branch from top to bottom successively decreases to 1/4, 1/2, and 1/4 of the input. Taking the number of input channels 16 as an example, the output channels of the three branches are 4, 8, and 4, in turn. Afterwards, smoothing is performed by a 1 × 1 convolution, and the output is added to the original input by a residual join. We replace one 5 × 5 convolution with two successive 3 × 3 convolutions, drastically reducing the number of parameters, and enhancing its linear expression. The formula for the parameter number M of the convolution layer can be written as
M = K × K × C1 × C2,(7)
where K is the size of the convolution kernel, C1 and C2 are the number of input and output channels. For example, the number of both input and output channels is C. The number of parameters for a convolution kernel of 5 × 5 can be written as
$$M1=5\;\times \;5\;\times \;C\;\times \;C=25C^{2}$$

In addition, the parameters with two convolution kernels of 3 × 3 can be written as
$$M2=3\;\times \;3\;\times \;C\;\times \;C+3\;\times \;3\;\times \;C\;\times \;C=18C^{2}$$

Then, the final output $F_{out}$ of the MFF module can be written as
(8)$$F_{out}={Conv}_{1\times 1}\;[\;C\;(F_{1},F_{2},F_{3})]+F_{in}$$
where ${Conv}_{1\times 1}$ is 1 × 1 convolution, C is the splicing of channel numbers, and $F_{in}$ is the input feature map.

### 3.4. Loss Function

This paper selects Soft Dice loss as the loss function. Soft Dice loss is a measure of overlap between two samples, which is suitable for binary image segmentation and can somewhat alleviate the quantity imbalance between positive and negative samples.
(9)$$Dice\;=\;\frac{2\sum_{i=1}^{n}x_{i}\cdot y_{i}+\varepsilon }{\sum_{i=1}^{n}x_{i}+\sum_{i=1}^{n}y_{i}+\varepsilon },$$
(10)$$L_{Dice}\;=\;-Ln\left(Dice\right)$$

$x_{i}$∈{0,1}, $y_{i}$∈{0,1} denote the region and ground truth of the model segmentation, respectively. ε is a minimal value, taken as 10^−5^, to avoid a denominator of 0; n = H × W denotes the number of pixels.

### 3.5. Experimental Settings and Evaluation Criteria

Our model is implemented on the Pytorch framework. Adaptive moment estimation (Adam) was selected for the optimizer, and Soft Dice loss was selected for the loss function. We set the initial learning rate as 0.0001, weight decay as $10^{-8}$, batch size as 16, and iteration epochs as 300. After 256 iterations, the learning rate decreases by 0.5. After the first convolution block, the number of input channels becomes 16. All of our training was implemented on an NVIDIA GeForce GTX 3090 GPU. We perform validation for each completed training round and use the model parameters that perform best on the validation set for the test set for one round.

Four metrics are used to estimate the performance of the models: Dice score, Average symmetric surface distance (ASSD), IoU, and Precision.
(11)$$Dice\;Score\;=\;\frac{2\sum_{i=1}^{n}x_{i}\cdot y_{i}}{\sum_{i=1}^{n}x_{i}+\sum_{i=1}^{n}y_{i}},$$
(12)$$IoU\;=\;\frac{\sum_{i=1}^{n}x_{i}\cdot y_{i}}{\sum_{i=1}^{n}x_{i}+\sum_{i=1}^{n}y_{i}-\sum_{i=1}^{n}x_{i}\cdot y_{i}}.$$

$x_{i}$∈{0,1}, $y_{i}$∈{0,1} denote the region and ground truth of the model segmentation, respectively. n = H × W denotes the number of pixels.
(13)$$ASSD\;=\;\frac{1}{\left|S_{x}|+|S_{y}\right|}\times \left(\sum_{x\in S_{x}}d\left(x,S_{x}\right)+\sum_{y\in S_{y}}d\left(y,S_{y}\right)\right).$$

$S_{x}$ and $S_{y}$, respectively, represent the boundary point set of model segmentation and the boundary point set of ground $d\left(n,S_{x}\right)={min}_{m\in S_{x}}(||n-m||)$￼ is the minimum Euclidean distance from n to all the points.
(14)$$Precision\;=\;\frac{TP}{TP\;+\;FP}.$$

Among them, TP (true positive) and FP (false positive) represent correctly segmented skin lesion pixels and background pixels incorrectly labelled as skin lesion pixels, respectively.

### 3.6. Data Pre-Processing

The dataset for this paper is taken from a public challenge: Lesion Boundary Segmentation in ISIC-2018 [25]. The challenge dataset comes from the ISIC-2017 dataset [26] and the HAM10000 dataset [27]. It can be found at the following website: https://challenge2018.isic-archive.com/task1. This dataset provides 2594 dermoscopic images with ground truth segmentation masks for training. At the same time, provide 100 and 1000 images without ground truth masks for validation and test set, respectively. In this paper, 2594 images with ground truth masks are used as a dataset because our study belongs to a fully annotated task. These images are randomly divided into 1816, 260, and 518 images in the ratio of 7:1:2 as the training set, validation set, and test set, respectively. All the images are adjusted to 256 × 342 because input image sizes vary from 771 × 750 to 9748 × 4499. The image size was randomly cropped to 224 × 300, and then the cropped image was randomly rotated in the horizontal or vertical direction at an angle of (−30°, 30°).

## 4. Experiments

This paper validates the effectiveness of MSF-Net by segmenting binary skin lesions from dermoscopic images, and the specific experimental results will be provided in this section.

### 4.1. Ablation Study

To demonstrate the validity of each module in MSF-Net, a series of ablation experiments were conducted to compare the performance of different modules. CA-Net is used as the baseline, a traditional five-tier structure. In order to prevent overfitting, we use Dropout on the last two layers of the network to provide a better generalization ability of the model. In our ablation experiment, all subjects were run in the same network environment, parameters, and data set.

We obtained eight comparison methods by separately adding and free-combining S-conv, MFF, and MDC modules. Eight methods with different module configurations were used to segment the ISIC2018 dataset, and the visual comparison of five representative methods is shown in Figure 6. It can be seen from Figure 6, when dealing with segmentation tasks with distracts (as shown in lines 1 and 2), compared with the Baseline method (P3), Baseline+S-conv (P4), Baseline+MFF (P5), and Baseline+MDC (P6) were more accurate in locating the lesion area. In contrast, the Baseline method incorrectly segmented some disturbance areas. When dealing with small-scale tasks (as shown in row 5), Baseline+S-conv adds a spatial attention mechanism that captures the target region more effectively. However, the Baseline+MFF could be more robust in segmentation because it aggregates large-scale spatial information and is more suitable for processing large-scale target tasks. However, all of them are better than the Baseline method. By integrating these methods, MSF-Net (P7) achieves the optimal segmentation effect. When dealing with tasks with complex boundary information (such as the picture in line 6), Baseline+MFF and Baseline+MDC fuse relevant information of multiple scales while adaptively selecting the perceptual field size of feature maps at different scales, making the segmentation much more effective.

MSF-Net achieved the best performance on the ISIC2018 dataset compared to adding a single module on the baseline (CA-Net). The performance indexes of several models are shown in Table 1. It can be seen from Table 1 that compared with baseline, the Dice indicators of the baseline+S-conv method, baseline+MFF method, and baseline+MDC method in the ISIC 2018 dataset increased by 0.45, 0.46, and 0.53%, the IoU indicators increased by 0.74, 0.79, and 0.9%, the Precision indicators increased by 2.62, 2.03, and 2.42%, and the ASSD indicators decreased by 0.0775, 0.0479, and 0.0855, respectively. MSF-Net achieves optimal performance on the ISIC2018 dataset by integrating three modules simultaneously. Dice, IoU, and Precision are 0.87, 1.45, and 3.89% higher than baseline, respectively, and ASSD indicators are 0.1157 lower, which proves the effectiveness of MSF-Net and each module.

To select a suitable combination of dilated convolution, we also conduct further ablation experiments. Five methods were obtained based on combining dilated convolutions with different expansion rates (r). The evaluation indexes of several methods are shown in Table 2. As can be seen from it, most of our evaluation indexes (Dice, et al. [5,28].) achieved the best results with the dilation rate (r = 1, 2, 4, 10), while ASSD indexes achieved the best results with the dilation rate (r = 1, 2, 4, 8). We choose the dilation rate (r = 1, 2, 4, 10) as the final dilated convolution combination.

In addition, the number of parameters increased by each module compared with the baseline, and the final model parameters are shown in Table 3. The parameter quantity of the S-conv module is similar to the baseline. The MFF module and MDC module have only 4% and 20% more parameters, respectively, than the baseline.

### 4.2. Results

The performance indicators of MSF-Net and 11 comparison methods on the ISIC2018 Dataset are quantitatively presented in Table 4, including six general segmentation networks and five networks dedicated to medical image segmentation, especially skin disease segmentation. In our comparison experiment, all competitors were run under the same computing environment, data set, and data enhancement to ensure a fair comparison. It can be seen from Table 4 that MSF-Net is the par excellence network in terms of all indicators and parameters. In addition, among all networks, denseASPP is the par excellence model in the universal segmentation model, while CPF-Net is the par excellence model for skin lesions segmentation. Among them, CPF-Net is the best performer in the Dice and IoU indices and is not far behind our network. MSF-Net showed the best results for Precision and ASSD. Meanwhile, the parameter of CA-Net is only 2.7M, which is the minimum network among all networks. In comparison, MSF-Net only has 0.67M more parameters than CA-Net, which is equivalent to 1/10 and 1/8 of denseASPP and CE-Net parameters. These indicators show that MSF-Net can acquire better segmentation results with fewer parameters. Therefore, MSF-Net is an excellent network.

Figure 7 visualizes the segmentation result of eight representative networks on the ISIC2018 dataset for visual comparison. These samples in Figure 7 contain a variety of complex segmentation tasks, some with excessive scale disparity, some with complex segmentation boundaries, some with low contrast, and some with blurred segmentation boundaries. As can be seen from Figure 7, MSF-Net outperforms other methods in general, and the segmentation results are closest to the ground truth. Row 5 in Figure 7 displays the segmentation results of these models for lesion images with small scales, blurred segmentation boundaries, and low contrast. The sixth row in Figure 7 shows the segmentation results of large-scale lesion images by different networks. Due to the lack of guidance on global context information, U-Net(p4) and AttU-Net(p5) have poor segmentation effects when segmenting these types of images. In contrast, MSF-Net extracts feature with crucial information by adding spatial attention to the convolution and capture local features and global context information through MDC and MFF, achieving a better segmentation effect. Rows 1, 2, and 3 in Figure 7 display the segmentation results for lesion images with blurred boundaries. CPFNet (p9) extracts information at different scales using a pyramid structure, which improves segmentation for low-contrast tasks but may still produce erroneous segmentation for complex samples with complex boundaries. On the contrary, thanks to MDC and MFF, MSF-Net (p11) has an advantage in processing samples with complex boundaries, and the segmented boundary information is relatively straightforward. The sixth row in Figure 7 shows the segmentation results of large-scale lesion images by different networks.

## 5. Discussion

In the comparison method shown in Table 4, DeeplabV3+ [24] introduces a large number of parallel expansion convolutions in the encoder part to extract features of different scales by setting different expansion rates. DenseASPP [29] combines ASPP in the DeepLab series with dense connections in denseNet. It uses dense connections to combine the output of each dilated convolution, so it has a larger receptive field and more dense sampling points. CE-Net [31] combined the idea of Inception-ResNet-V2 and dilated convolution and proposed a dilated convolution block (DAC) with four parallel branches to encode high-level semantic feature mapping. The fields of the four branches are 3, 7, 9, and 19, respectively. Compared with the three multi-scale methods above, MSF-Net has two significant advantages: First, the MDC module can adaptively adjust its receptive field size and extract multi-scale context information effectively. In the MDC module (as shown in Figure 4), considering the feature correlation between adjacent scales, we fuse the feature graphs output by two neighboring dilated convolution branches. The spatial attention mechanism generates two attention coefficient maps and then acts on different feature maps, respectively, so that the two feature maps are combined with different weight coefficients, and it is possible to generate different sizes of perceptual fields for different attention points of these branches, thus realizing adaptive adjustment of their perceptual field sizes according to the input. Then, the parameters in MDC and MFF modules are relatively small because they use a large number of 1 × 1 convolutions in both modules to reduce the number of channels. In addition, MSF-Net only needs to down the number of image channels to 256 to achieve a good segmentation effect. At the same time, weights are shared among our dilated convolution branches, where a larger receptive field is obtained without adding additional parameters. However, the parameters of these three multi-scale methods approaches or exceed the traditional convolution, and the number of down-sampled channels is much larger than that of MSF-Net.

Compared with some of the best current networks in the field of dermatological segmentation, such as CPF-Net [32], MSF-Net has Dice, and IoU metrics similar to it, but Precision and ASSD metrics are better than CPF-Net, and the number of parameters in MSF-Net is less than one-tenth of CPF-Net. The parameter quantity of Ms-Red [28] is similar to MSF-Net, but all four evaluation indicators are inferior to MSF-Net. However, compared to some networks with simple structures, such as U-net [12], our model is more complex, which slightly affects the processing speed of the network. Further optimization of the model is needed in the following work to improve the processing speed of the network. In addition, our network belongs to a fully annotated segmentation task, where the dataset must carry a ground truth mask and is insufficient to identify the type of lesion for the segmented lesion region. Our subsequent research can be divided into two directions. The first is to handle weak and semi annotation tasks, and the second is to recognize and classify segmentation results.

Image segmentation belongs to an important branch in computer-aided diagnosis, and our network has certain advantages in terms of parameters and segmentation results compared with some excellent networks in the current field of dermatology segmentation. In the future, it can be combined with image classification to achieve a computer-aided diagnosis of skin diseases without human intervention.

## 6. Conclusions

This paper proposes a deep neural network (MSF-Net) for skin lesion segmentation based on CA-Net, for some problematic segmentation tasks, especially lesion images with excessive scale disparity, irregular shape of the lesion area, low contrast with the background, and blurred borders. We propose three core modules such as S-Conv, MDC, and MFF. Specifically, MSF-Net introduces a spatial attention mechanism in ordinary convolutional blocks to focus on crucial locations, suppress irrelevant information, and localize lesion regions. Meanwhile, MDC and MFF modules are introduced after the convolutional layer to extract contextual information at different scales and adaptively adjust the perceptual field size of the feature map. We conducted many experiments on the publicly available dataset ISIC2018 to evaluate the performance of MSF-Net. In the ablation experiments, all three of our modules outperformed CA-Net regarding performance metrics. In the comparison experiments, we compared eleven existing state-of-the-art models. The experimental results show that MSF-Net can obtain relatively good segmentation result with smaller parameters. In addition, the shape of the output feature map of each module is the same as the input, which is highly flexible and can be applied well in other areas of medical images.

## Figures and Tables

**Figure 1 biomedicines-11-01733-f001:**
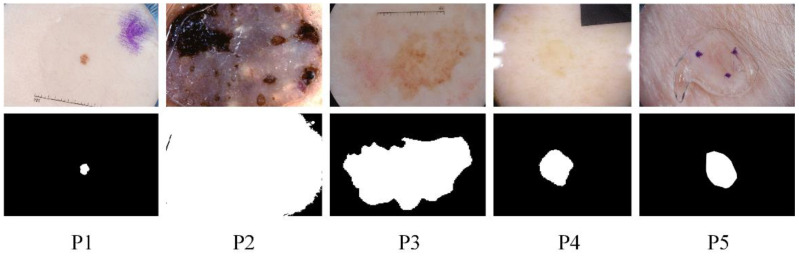
Some examples of challenging skin lesions in the public dataset ISIC 2018 include small-scale lesions (**P1**), large-scale lesions (**P2**), irregularly shaped lesions (**P3**), lesions with low contrast to the background (**P4**), and lesions with blurred boundaries (**P5**). The top is the lesion picture, and the bottom is the corresponding ground truth.

**Figure 2 biomedicines-11-01733-f002:**
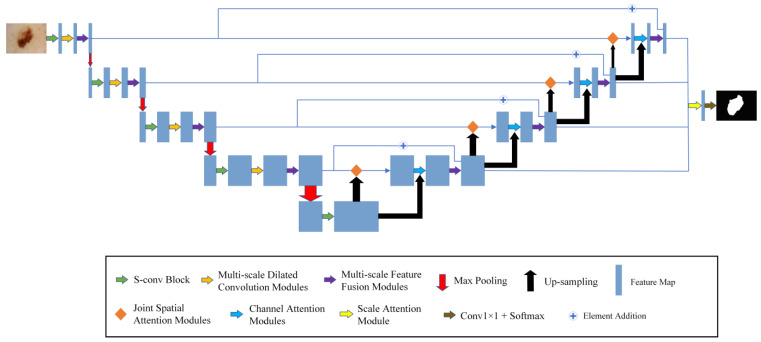
The general framework of MSF-Net.

**Figure 3 biomedicines-11-01733-f003:**
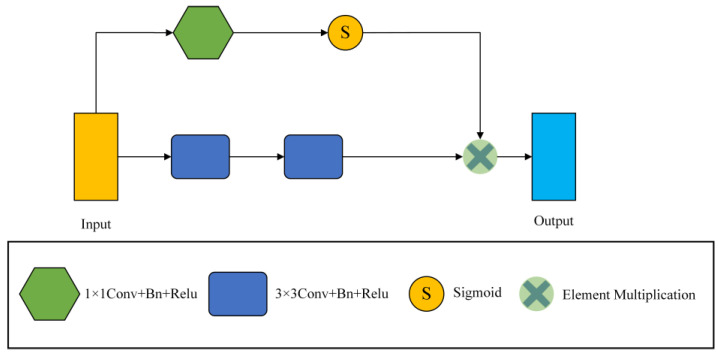
The general framework of the S-conv Block.

**Figure 4 biomedicines-11-01733-f004:**
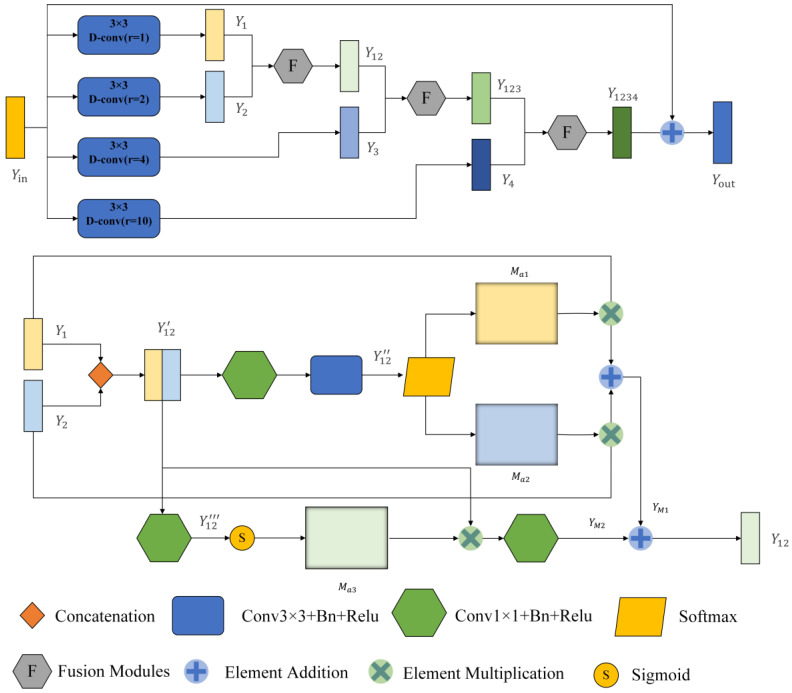
General framework of the Multi-scale Dilated Convolution Modules (MDC). The top is a multi-scale feature extraction module, and the bottom is a feature fusion module. After dilated convolution, rectangular boxes with different colors represent the output feature graph.

**Figure 5 biomedicines-11-01733-f005:**
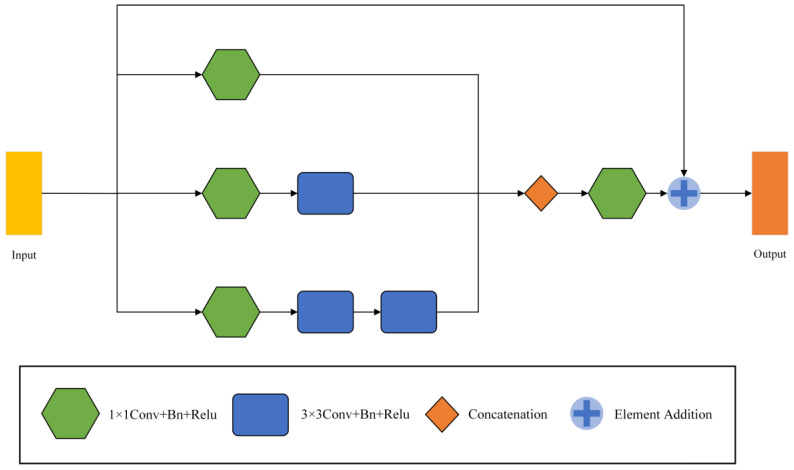
The framework of Multi-scale Feature Fusion Modules (MFF).

**Figure 6 biomedicines-11-01733-f006:**
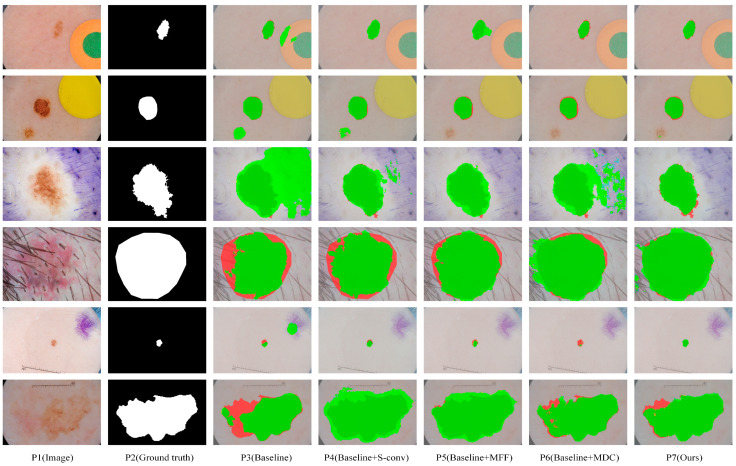
The segmentation results of our ablation experiments were visually compared on the ISIC2018 dataset. Among them, dark green, red, and green indicate correct segmentation, under-segmentation, and over-segmentation, respectively.

**Figure 7 biomedicines-11-01733-f007:**
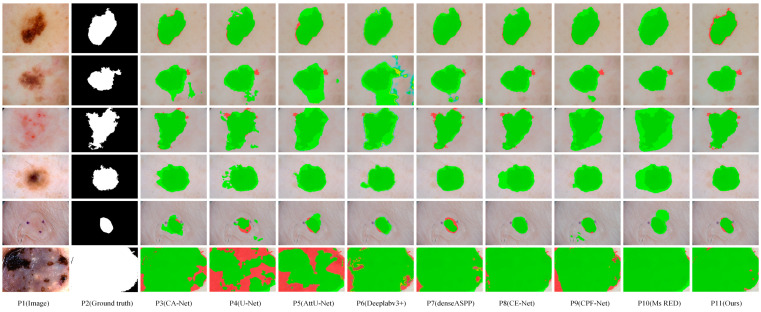
Visual comparison of skin lesion segmentation results of different models on the ISIC2018 dataset, where dark green, red, and green indicate correct segmentation, under-segmentation, and over-segmentation, respectively.

**Table 1 biomedicines-11-01733-t001:** Statistical comparison of ablation experiments for three modules in MSF-Net. Among evaluation indexes, the lower ASSD indicator proves that the network is better, while the other indicators are the opposite.

Method	Dice (%)	IoU (%)	Precision (%)	ASSD (pix)
Baseline	93.02 ± 0.50	87.30 ± 0.88	88.28 ± 0.32	0.4645 ± 0.0163
Baseline+S-conv	93.47 ± 0.57	88.04 ± 0.96	90.90 ± 0.86	0.3870 ± 0.0216
Baseline+MFF	93.48 ± 0.44	88.09 ± 0.71	90.31 ± 1.06	0.4166 ± 0.0560
Baseline+MDC	93.55 ± 0.39	88.20 ± 0.63	90.70 ± 0.71	0.3760 ± 0.0091
Baseline+S-conv+MFF	93.44 ± 0.29	88.00 ± 0.48	90.38 ± 0.20	0.3966 ± 0.0140
Baseline+S-conv+MDC	93.91 ± 0.28	88.77 ± 0.46	90.85 ± 0.31	0.3635 ± 0.0046
Baseline+MFF+MDC	93.60 ± 0.37	88.27 ± 0.62	90.65 ± 0.49	0.3767 ± 0.0234
Baseline+S-conv+MFF+MDC	93.89 ± 0.47	88.75 ± 0.77	92.17 ± 0.07	0.3488 ± 0.0172

**Table 2 biomedicines-11-01733-t002:** Statistical comparison of ablation experiments of MDC modules with different cavity rate combinations.

Method	Dice	IoU	Precision	ASSD
Baseline+MDC (r = 1, 2, 4)	0.9378	0.8857	0.8887	0.3847
Baseline+MDC (r = 1, 2, 6)	0.9358	0.8824	0.8962	0.3937
Baseline+MDC (r = 1, 2, 4, 8)	0.9384	0.8866	0.8978	0.3638
Baseline+MDC (r = 1, 2, 4, 10)	0.9393	0.8882	0.8999	0.3669
Baseline+MDC (r = 1, 2, 6, 10)	0.9367	0.8838	0.8807	0.3741

**Table 3 biomedicines-11-01733-t003:** The increase in the number of participants per module in our network is compared to the baseline and the final number of participants for our approach.

Modules	Baseline	S-conv	MFF	MDC	Ours
Params(M)	2.7884	0.0002	0.1191	0.5519	3.4596

**Table 4 biomedicines-11-01733-t004:** Skin lesion segmentation performance of different networks on ISIC2018. The backbones of Deeplabv3+ and DenseASPP are ResNet50 and DenseNet161, respectively.

Networks	Parameters (M)	Dice (%)	IoU (%)	Precision (%)	ASSD (pix)
CA-Net [5]	2.79	93.02 ± 0.50	87.30 ± 0.88	88.28 ± 0.32	0.4645 ± 0.0163
U-Net [12]	34.53	93.12 ± 0.30	87.45 ± 0.52	88.79 ± 0.14	0.4439 ± 0.0261
AttU-Net [13]	34.88	93.05 ± 0.11	87.34 ± 0.17	88.75 ± 0.70	0.4203 ± 0.0199
Unet++ [14]	36.63	93.08 ± 0.50	87.43 ± 0.79	89.60 ± 1.11	0.5100 ± 0.0783
R2UNet [16]	39.09	88.81 ± 1.23	80.61 ± 1.90	80.02 ± 1.74	0.8013 ± 0.0837
Deeplabv3+ [24]	39.76	93.60 ± 0.37	88.29 ± 0.60	90.09 ± 0.26	0.3905 ± 0.0185
denseASPP [29]	35.37	93.76 ± 0.39	88.53 ± 0.63	89.84 ± 1.16	0.3778 ± 0.0232
BCDU-Net [30]	28.80	92.96 ± 0.38	87.19 ± 0.60	90.19 ± 0.16	0.4566 ± 0.0132
CE-Net [31]	28.98	93.93 ± 0.68	88.84 ± 1.13	90.11 ± 0.89	0.3850 ± 0.0428
CPF-Net [32]	43.25	93.95 ± 0.44	88.86 ± 0.72	90.30 ± 0.56	0.3720 ± 0.0227
Ms RED [28]	4.02	93.59 ± 0.28	88.27 ± 0.44	90.66 ± 0.68	0.4093 ± 0.0002
MSF-Net (Ours)	3.46	93.89 ± 0.47	88.75 ± 0.77	92.17 ± 0.07	0.3488 ± 0.0172

## Data Availability

The data presented in this study are available from the corresponding author on reasonable request.

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
