# Peer review of "MSF-Net: A Lightweight Multi-Scale Feature Fusion Network for Skin Lesion Segmentation"

_biomedicines, 2023, doi:10.3390/biomedicines11061733_

Round 1
Reviewer 1 Report
The paper by Shao et al describes applicaton of novel multi‐scale feature fusion network for skin lesion segmentation. The theme of the study is important and interesting, and the authors fully describe the proposed approaches. At the same time there is a lack of information regarding analyzed dataset and in general some major issues must be eliminated prior to publication:
1. Dataset of 2594 images is presented in the beginnong of Results section. Please, shift this part of the text to the Methods and fully describe the dataset (with references).
2. Section 4.2 belongs to the Methods section too.
3. Separation of data into train, validation and test descbed very poorly. Additional explanations are required. The authors just separated some part of the images into test set, but how this part of the images was selected? Randomly? What if some other part of the images will be selected for the test set? How will change the perormance of the model?
4. Please, provide metrics of the model (eg as in Table 1) for training, validation and test.
5. What is presented in green in figures 6 and 7? please, update the figures legends.
6. Discussion section is missing references. Add numeral references to the studies in the field and compare the obtained results with results of other studies.
The paper may be published only after correction of mentioned issues.
Reviewer 2 Report
The paper "MSF-Net: A Lightweight Multi-Scale Feature Fusion Network for Skin Lesion Segmentation" investigates the efficiency for skin lesion segmentation of a multi‐scale feature fusion network based on comprehensive attention convolutional neural network, in comparison with different convolution models. The valuable aim is to propose an effective supportive mean in the early diagnosis of skin lesions and melanoma. English is good, the frameworks of convolution models are clearly illustrated, the comprehensive work is well organized and results convincing. Only some points need attention, as detailed to the Authors, to make the paper suitable for publication.
In general, it should be clearly reported the classification of the lesions used to investigate the skin lesion segmentation ability of the multi‐scale feature fusion network as compared with the different modules. The kind of lesions analyzed could be indicated, for example, in the legend of Figure 1.
A pathological classification of the lesions investigated (lines 255-261) should be of value also to introduce and discuss possible limitations to this study, and the perspectives of future work for the effective diagnosis from the segmentation of skin pathological pictures.
Also, at the convenience of less specialized readers, the meaning of “ground truth” should be explained.
English is good
Round 2
Reviewer 1 Report
The authors addressed all arised issues. The paper may be published.
The only shortcoming is a symbol of "second" that should appear as "2nd" but the authors utilize "2st".
